# Metallo-sideromycin as a dual functional complex for combating antimicrobial resistance

Chenyuan Wang [1,6], Yushan Xia[1,6], Runming Wang[1], Jingru Li[1,2], Chun-Lung Chan[1], Richard Yi-Tsun Kao[3], Patrick H. Toy[1], Pak-Leung Ho [3,4], Hongyan Li [1,5] ✉ & Hongzhe Sun [1,5] ✉

The rapid emergence of antimicrobial resistance (AMR) pathogens highlights the urgent need to approach this global burden with alternative strategies. Cefiderocol (Fetroja®) is a clinically-used sideromycin, that is utilized for the treatment of severe drug-resistant infections, caused by Gram-negative bacteria; there is evidence of cefiderocol-resistance occurring in bacterial strains however. To increase the efficacy and extend the life-span of sideromycins, we demonstrate strong synergisms between cefiderocol and metallodrugs (e.g., colloidal bismuth citrate (CBS)), against *Pseudomonas aeruginosa* and *Burkholderia cepacia*. Moreover, CBS enhances cefiderocol efficacy against biofilm formation, suppresses the resistance development in *P. aeruginosa* and resensitizes clinically isolated resistant *P. aeruginosa* to cefiderocol. Notably, the co-therapy of CBS and cefiderocol significantly increases the survival rate of mice and decreases bacterial loads in the lung in a murine acute pneumonia model. The observed phenomena are partially attributable to the competitive binding of $Bi^{3+}$ to cefiderocol with $Fe^{3+}$, leading to enhanced uptake of $Bi^{3+}$ and reduced levels of $Fe^{3+}$ in cells. Our studies provide insight into the antimicrobial potential of metallo-sideromycins.

Multidrug resistant (MDR) bacterial infections are becoming a serious threat to human health worldwide. The lack of new clinically approved antibiotics in recent decades and the overuse/misuse of known antibiotics have accelerated the development of drug resistance in bacteria[1,2]. In particular, Gram-negative bacterial infections are particularly prevalent and difficult to treat due to their complicated structures compared to Gram-positive bacteria, and their highly active efflux pump systems, which both interfere with antibiotic enrichment in bacterial cells[3]. For example, *P. aeruginosa*, one of the ESKAPE pathogens highlighted by the Infectious Disease Society of America (IDSA)[4], is now a major threat in burned patients[5]

and immunocompromised patients[6]. The high level of intrinsic resistance of *P. aeruginosa* to known antibiotics is partially due to their outer membrane (OM) permeability and the expression of efflux pumps, both of which can prevent the accumulation of antibiotics at the bacterial target sites[7,8]. Therefore, there is an urgent need to either develop new types of antibiotics or improve the efficacy of clinically used antibiotics by alternative strategies.

One strategy to increase the antibacterial activity of known antibiotics is the Trojan Horse strategy, by which drugs can be delivered into the bacterial cells through nutrient uptake pathways to bypass the membrane barrier. Siderophores are frequently used in this strategy

[1]Department of Chemistry, The University of Hong Kong, Pokfulam Road, Hong Kong SAR, PR China. [2]Chemistry and Chemical Engineering Guangdong Laboratory, Guangdong, PR China. [3]Department of Microbiology, The University of Hong Kong, Sassoon Road, Hong Kong SAR, PR China. [4]Carol Yu Centre for Infection, The University of Hong Kong, Sassoon Road, Hong Kong SAR, PR China. [5]State Key Laboratory of Synthetic Chemistry and CAS-HKU Joint Laboratory of Metallomics for Health and Environment, The University of Hong Kong, Pokfulam Road, Hong Kong SAR, PR China. [6]These authors contributed equally: Chenyuan Wang, Yushan Xia. ✉e-mail: hylichem@hku.hk; hsun@hku.hk

since they are produced by microorganisms to uptake ferric iron ($Fe^{3+}$) from the host environment actively[9]. Siderophore-drug conjugates, named as sideromycins, are a group of antibiotics that can be uptake into the bacterial cells through active transport pathways[10–12]. Interestingly, sideromycins are first found as naturally occurring antibiotics. For example, albomycins were first obtained from *Streptomycetes* strains[13], characterized by Benz et al.[14] in 1982, and used successfully in people in the late 1940s against penicillin-resistant bacteria[13]. Albomycins consist of a trihydroxamate ferrichrome-like siderophore, which can be recognized and utilized as ferrichrome through TonB transporters. After being internalized, the thioribosyl-pyrimidine warhead as a seryl t-RNA inhibitor is released by peptidization and inhibits protein synthesis. Thus albomycins showed a 100-fold lower minimum inhibitory concentrations (MICs) than those antibiotics that enter cells by passive diffusion, and exhibited in vivo antimicrobial activity in a mouse infection model[15]. Inspired by these natural sideromycins, semi- and total-synthesized sideromycins with different siderophores, drugs/warheads, and linkers were designed to fight against AMR either as narrow or broad spectrum antibiotics[16–18]. Despite numerous attempts, until now only one sideromycin, namely cefiderocol (S-649266, Fetroja®), consisting of a parenteral cephalosporin named ceftazidime and a chlorocatechol moiety as a mimic of the natural bis-catechol components of natural siderophore at the third-position in the side chain (Fig. 1a)[19], was approved for medical use in 2019 in the United States to treat complicated urinary tract infections. The indications were subsequently broadened to hospital-acquired bacterial pneumonia and ventilator-associated bacterial pneumonia. However, the emergence of cefiderocol-resistant *P. aeruginosa* and other Gram-negative bacterial strains has been reported recently by various

mechanisms including mutations of siderophore receptors and increased expression of efflux pumps[20,21]. Therefore, alternative strategies are pressingly needed to prolong the life-span of this type of antibiotics.

Metal compounds have been historically utilized as antimicrobial agents[22–24]. Owing to their multi-targeted modes of action[25], metal compounds have received growing attention for tackling antimicrobial resistance[26,27], and in fact are being recognized as a promising source of antibiotics[28]. In particular, bismuth(III) and gallium(III) compounds have been shown to either serve as antimicrobial agents or as resistance breakers to tackle antimicrobial resistance[29,30]. Previously, $Bi^{3+}$ was established to compete with $Fe^{3+}$ for its cellular uptake[31], and $Ga^{3+}$ is also known to mimic $Fe^{3+}$ in biological systems to disrupt $Fe^{3+}$ functions[32,33]. We thus hypothesize that simultaneous delivery of sideromycins and metals such as $Bi^{3+}/Ga^{3+}$ via a dual Trojan Horse strategy through metallo-sideromycins, might kill bacteria synergistically and also slow down the development of high-level resistance in bacteria.

Herein, we selected cefiderocol (named as CEF thereafter) as a showcase to validate such a strategy and screened a series of metallo-compounds against different bacterial strains. We found a strong synergy between $Bi^{3+}/Ga^{3+}$ compounds and CEF against *Pseudomonas* including clinically isolated CEF-resistant strains and *Burkholderia* strains. Notably, the metallo-sideromycin is relatively resistance-proof in comparison to sideromycin per se, implying the potential of extending the life-span of CEF. Importantly, the in vitro efficacy of the combination of a clinically used drug, CBS, and CEF could be well translated into in vivo. We further show that the reduced uptake of $Fe^{3+}$ owing to the competition of $Bi^{3+}$ via binding to CEF and enhanced

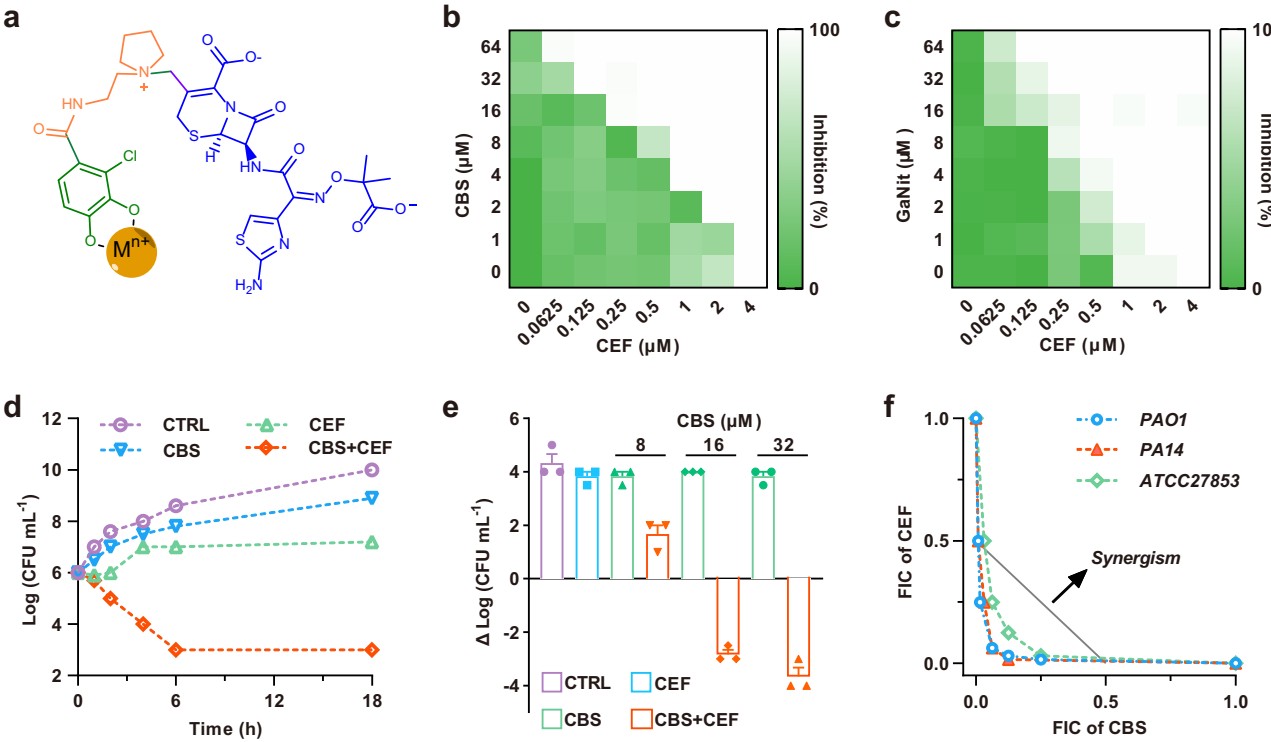

**Fig. 1 | Bismuth/gallium compounds have synergistic effects with cefiderocol against *P. aeruginosa*. a** Proposed chemical structure of a metallo-sideromycin using cefiderocol (CEF, S-649266) as an example. The antibiotic (ceftazidime), linker and siderophore (i.e., 3-chlorocatechol) moiety are highlighted in purple, orange and green respectively. Representative heat plots of microdilution checkboard assay for the combination of CEF and CBS (**b**) or GaNit (**c**). **d** Time killing curves for CBS and CEF individually and in combination against PAO1 during 24 h incubation. The concentration of CBS and CEF are 16 and 0.5 μM, respectively. **e** Bar chart showing bacterial loads exposed to the CBS and CEF combination or individually. The concentration of CEF is 0.5 μM. $n = 3$ biologically independent samples. Mean value of three replicates are shown and error bars indicate ±SEM. **f** Isobolograms of the combination of CBS and CEF against different *P. aeruginosa* strains. The gray dash line represents the ideal isobole at which drugs are additively and independently. Data points below this line are defined as synergy. CTRL control, CEF cefiderocol, CBS colloidal bismuth citrate, GaNit gallium nitrate, FIC fractional inhibitory concentration. Source data are provided as a Source Data file.

**Table 1 | Antibacterial activity of CEF in the absence and presence of different metal compounds against different bacteria**

| Strains | MIC of CEF (µM) | | | | | | | |
|---|---|---|---|---|---|---|---|---|
| | In the presence of 50 µM of metal compounds | | | | | | | |
| | Alone | $Bi^{3+}$ | $Ga^{3+}$ | $Co^{3+}$ | $Cr^{3+}$ | $Fe^{3+}$ | $Mn^{2+}$ | $Ti^{4+}$ |
| *P. aeruginosa* PAO1 | 4 | 0.06 | 0.06 | 4 | 4 | 4 | 8 | 4 |
| *P. aeruginosa* PA14 | 4 | 0.12 | 0.06 | 4 | 4 | 8 | 8 | 4 |
| *P. aeruginosa* ATCC27853 | 2 | 0.12 | 0.12 | 4 | 4 | 4 | 4 | 4 |
| *B. cepacia* ATCC25416 | 2 | 0.03 | 0.06 | 4 | 2 | 2 | 4 | 2 |
| *B. cepacia* J2315 | 0.5 | 0.03 | 0.03 | 4 | 0.5 | 0.5 | 0.5 | 0.5 |
| *E. coli* ATCC25922 | 1 | 1 | 1 | 1 | 1 | 2 | 2 | 1 |
| *E. coli* ATCC10536 | 0.03 | 0.03 | 0.03 | 0.125 | 0.03 | 0.06 | 0.06 | 0.03 |
| *E. coli* (NDM-1)⁺ | 1 | 1 | 1 | 2 | 1 | 2 | 4 | 2 |
| *E. coli* clinical | 1 | 1 | 1 | 1 | 1 | 1 | 1 | 1 |

Metal compounds used are: $Bi^{3+}$—colloidal bismuth citrate (CBS); $Ga^{3+}$—Ga(NO₃)₃ (or GaNit); $Co^{3+}$—Co(OAc)₃; $Cr^{3+}$—Cr₂(SO₄)₃; $Fe^{3+}$—FeCl₃; $Mn^{2+}$—Mn(OAc)₂; $Ti^{4+}$—Ti(IV)-citrate.
*CEF* cefiderocol, *MIC* minimum inhibitory concentration.

uptake of $Bi^{3+}$ are likely attributable to the observed phenomena. The role of metallo-sideromycins in the fight against AMR is discussed.

## Results

### Bismuth/gallium compounds and CEF synergistically inhibit *P. aeruginosa* growth

We first performed a primary screening to examine potential synergisms between metal compounds and CEF against Gram-negative bacterial strains including *Pseudomonas aeruginosa, Burkholderia cepecia, Escherichia coli, Klebsiella pneumoniae, Acinetobacter baumannii, Salmonella enterica, Enterobacter aerogenes, Aeromonas hydrophila, Vibrio cholerae* and *Proteus mirabilis*. The minimum inhibitory concentrations (MICs) of CEF against these bacterial strains were examined in the presence of 50 µM of metal compounds including colloidal bismuth citrate (CBS), Ga(NO₃)₃ (GaNit), FeCl₃, Ti(IV)-citrate, Mn(OAc)₂, Co(OAc)₃ and Cr₂(SO₄)₃ at different concentrations (half dilution from 2×MIC) of CEF after inoculation in cation-adjusted Mueller–Hinton broth (CA-MHB) culture medium for 24 h (Table 1 and Supplementary Table 1). Primary screening showed that the MICs of CEF could be reduced by 32–64 folds upon adding 50 µM CBS or GaNit against *P. aeruginosa* and *B. cepacia* strains. In contrast, the MICs of CEF remained almost unchanged upon the addition of other metal-containing compounds against all bacterial strains tested. The observed discrepancy in the effects of metal ions on CEF efficacy in different pathogens may be attributed to the fact that different bacterial strains have different iron requirements, uptake pathways, and different susceptibilities to $Bi^{3+}$/$Ga^{3+}$ compounds[34,35]. To validate this hypothesis, we also measured the intracellular bismuth concentration in *E.coli* (as an example) after being co-treated with CBS and CEF, and found similar bismuth concentrations in both *E.coli* and *P. aeruginosa* (Supplementary Fig. 1), indicating that similar amounts of bismuth were transported into the bacterial cells. However, CBS was shown to be more toxic to *P. aeruginosa* than to *E.coli*[34], thus the selective antimicrobial activity of $Bi^{3+}$/$Ga^{3+}$, which are delivered to specific bacterial cells through cefiderocol siderophore pathways, is also accountable for the selective toxicity of the combination therapy.

Given that CBS and GaNit exhibited the best performance in enhancing CEF antimicrobial efficacy, we further investigated the underlying mechanism. We first evaluated the interaction between bismuth/gallium compounds and CEF against *P. aeruginosa* by the standard checkboard microdilution method using PAO1 standard strains as a showcase. CBS itself showed no or minor growth inhibition of PAO1 even at 64 µM. In contrast, the combination of CBS with CEF resulted in reduced MICs of CEF to 0.06 µM in the $Fe^{3+}$ sufficient medium (CAMHB) (Fig. 1b) and 0.125 µM in the iron-deficient condition (M9 broth) (Supplementary Fig. 2). This suggests that $Fe^{3+}$ only has a

slight effect on synergism under normal nutrient medium concentration. The fractional inhibitory concentration indexes (FICIs) were determined to be 0.125 and 0.24 for the iron-poor and iron-sufficient conditions respectively, suggesting a strong synergistic interaction (FICI < 0.25) between CBS and CEF. Furthermore, compared with ceftazidime (the parent antibiotic) and 3-chlorocatechol (the siderophore mimic), CEF showed more enhanced antibacterial activity when cotreated with CBS, indicating that the 3-chlorocatechol moiety as a siderophore mimic plays an important role in the dual Trojan Horse strategy (Supplementary Fig. 3). Similar synergistic effect was observed for GaNit and CEF against PAO1 with a FICI of 0.312 (Fig. 1c). Given that CBS combined with CEF showed the best antibacterial activity against PAO1, we thus selected CBS as a showcase for further study. The synergy between CBS and CEF was further confirmed by time killing assay. The population of PAO1 at the exponential phase was lowered by more than 4 logs upon exposure to the drug combination of CEF and CBS for 24 h at concentrations of 0.5 and 16 µM, respectively, compared with that with CEF or CBS alone (Fig. 1d and Supplementary Fig. 4). Furthermore, the bacterial loads were dropped by CBS in a dose-dependent manner when combined with CEF (Fig. 1e). Similarly, bismuth nitrate (Bi(NO₃)₃), bismuth N-acetylcysteine (Bi(NAC)₃), and bismuth subsalicylate (BSS) also exhibited synergistic effects with CEF against PAO1 (Supplementary Fig. 5). Moreover, 60 clinical *P. aeruginosa* strains that are CEF sensitive were screened and 40% of them showed synergy after co-treated with cefiderocol and CBS (Supplementary Fig. 6 and Supplementary Table 2).

### CBS and CEF synergistically inhibit *P. aeruginosa* biofilm formation

Since biofilm formation is one of the main strategies for *Pseudomonas* strains to produce tolerance to antibiotics under stress conditions, we next assessed whether bismuth compounds (e.g., CBS) could synergistically inhibit *P. aeruginosa* biofilm formation with CEF. First, we determined the anti-biofilm activity of CEF in the absence or presence of CBS by crystal violet method. As shown in Fig. 2a, both CBS and CEF showed limited anti-biofilm formation activities individually. The relative biofilm viability remains around 80% and 70% after the treatment with CEF and CBS at concentrations of 0.25 and 16 µM, respectively. On the contrary, the combination of CBS with CEF at subinhibitory concentrations showed distinct inhibition of biofilm formation and the relative biofilm viability decreased by more than 80%. In the meantime, the biofilm formation was also assessed by plate counting in the presence of CBS and CEF alone or in combination (Fig. 2b), which showed the same results as crystal violet assay and the bacterial loads were decreased to $10^3$ CFU/cm² under the combination treatment. Moreover, the biofilm formation was also quantified by

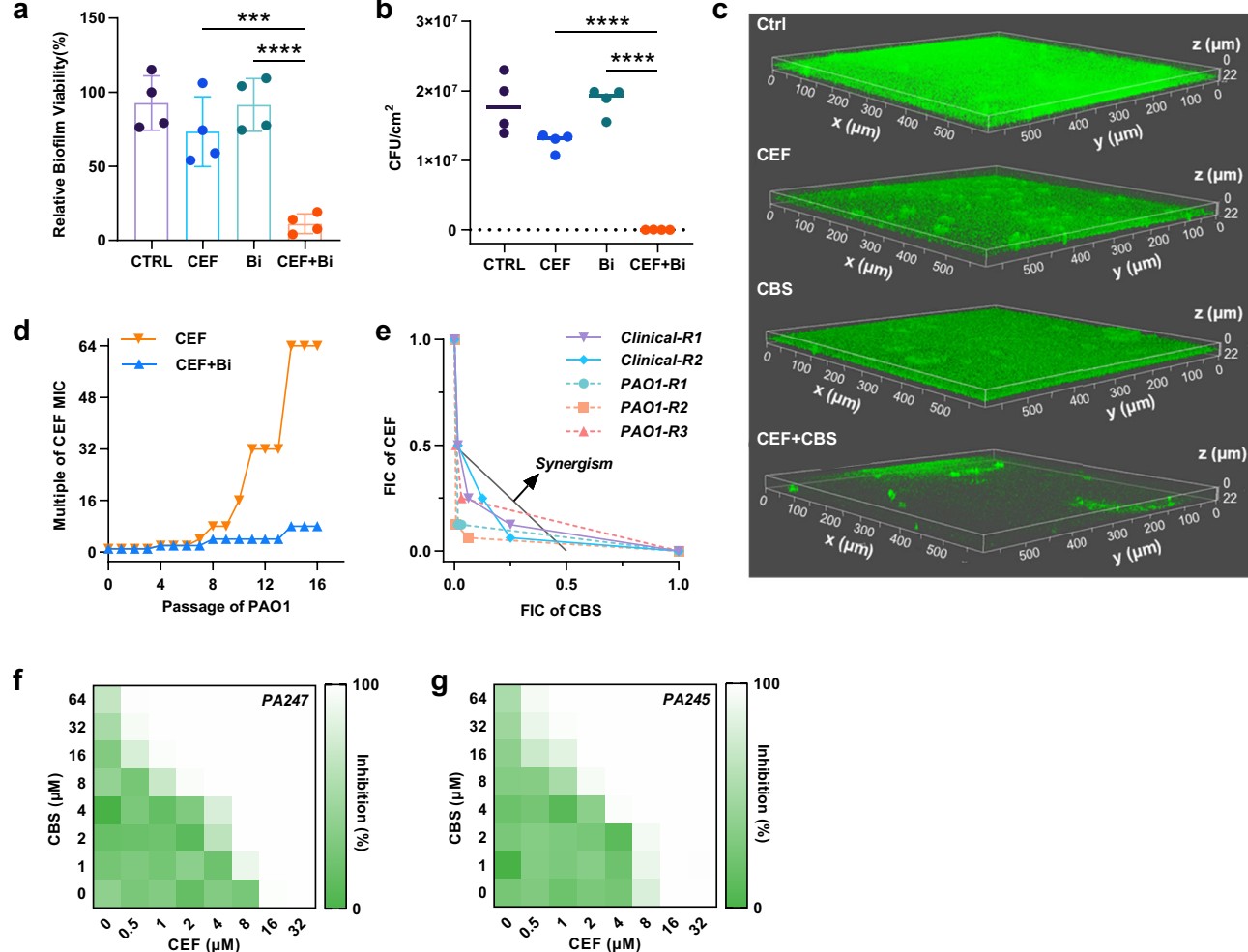

**Fig. 2 | CBS inhibits the formation of the biofilm and suppresses the evolution of CEF resistance.** Crystal violet assay (**a**) and plate counting of CFU (**b**) for anti-biofilm activity against PAO1 biofilm formation during 48 h incubation. n = 4 biologically independent samples. Mean value of four replicates are shown and error bars indicates ±SEM. p values were determined by one-way ANOVA test. ***p = 0.0009, ****p < 0.0001 means significant difference compared with control or single treated group. **c** Representative confocal laser scanning microscopy (CLSM) images of PAO1 biofilm formation in PAO1-GFP mutant strains with untreated control (CTRL); CBS (16 μM); CEF (0.25 μM) and combination of CBS (16 μM) and CEF (0.25 μM). Experiments were performed in triplicate and LB broth with no

drugs served as a control group. **d** Serial passage of PAO1 at sub-inhibitory concentration of CEF or the combination of CEF and CBS. The MIC test was performed every passage. **e** Isobolograms of the combination of CBS and CEF against laboratory-cultured CEF-resistant *P. aeruginosa* strains. The gray dash line represents the ideal isobole at which drugs are additively and independently. Data points below this line are defined as synergy. **f**, **g** Representative heat plots of micro-dilution checkboard assay for the combination of CEF and CBS against *PA247* and *PA245* clinical isolates. CTRL control, CEF cefiderocol, CBS colloidal bismuth citrate, FIC fractional inhibitory concentration. Source data are provided as a Source Data file.

confocal laser scanning microscopy. PAO1-GFP (Green Fluorescent Protein) fluorescent bacterial strain, produced by transferring *gfp* containing plasmid pEX18TC-gfp into PAO1 cells, was used for confocal imaging. As shown in Fig. 2c, the treatment of PAO1-GFP with CBS and CEF combination resulted in apparent decreases in the integrity and the thickness of biofilm compared to the untreated control group and single component groups.

**CBS suppresses the CEF resistance evolution in *P. aeruginosa***
Despite being used clinically for decades for the treatment of *Helicobacter pylori* infections, bismuth drugs are still effective and there appear no resistant bacterial strains to bismuth(III) drugs. The sustained efficacy of bismuth drugs is attributed to the multiple targeted mode of action[23,36]. On the contrary, the resistance to CEF in *P. aeruginosa* and *A. baumannii* has been reported recently[37]. One of the mechanisms of the reported resistance to CEF is the presence of various β-lactamases, for example, the metallo-β-lactamases e.g., NDM, which caused resistance to CEF in *E. coli* strains in vivo. Other

resistant mechanisms may include permeability defects and increased efflux, which may affect the siderophore transportation and accumulation, thus reducing the final concentration in the bacterial cells[21]. To investigate whether bismuth(III) compounds may exhibit the resistance-proof ability when combined with CEF against PAO1, mutant prevention concentration (MPC) was determined at different concentrations of CBS. Our results showed that CEF is unable to kill any high-level resistant mutants at 16-fold MIC and the MPC value was determined to be 32-fold MIC (Supplementary Fig. 7). Interestingly, CBS substantially reduced the MPC value of CEF in a concentration-dependent manner. The MPC to CEF was lowered to 2-fold MIC when co-treated with 128 μg ml⁻¹ CBS. In addition, PAO1 strain was performed serial passages in the presence of CEF or a combination of CEF and CBS over 16 days to assess the resistance-proof ability of CBS under long-term antibiotic stress. The MIC of CEF increased to 64-fold MIC (256 μM) after 16 passages under the treatment of CEF alone. In contrast, the MIC of CEF under combination treatment with CBS only raised to 2-fold MIC (Fig. 2d),

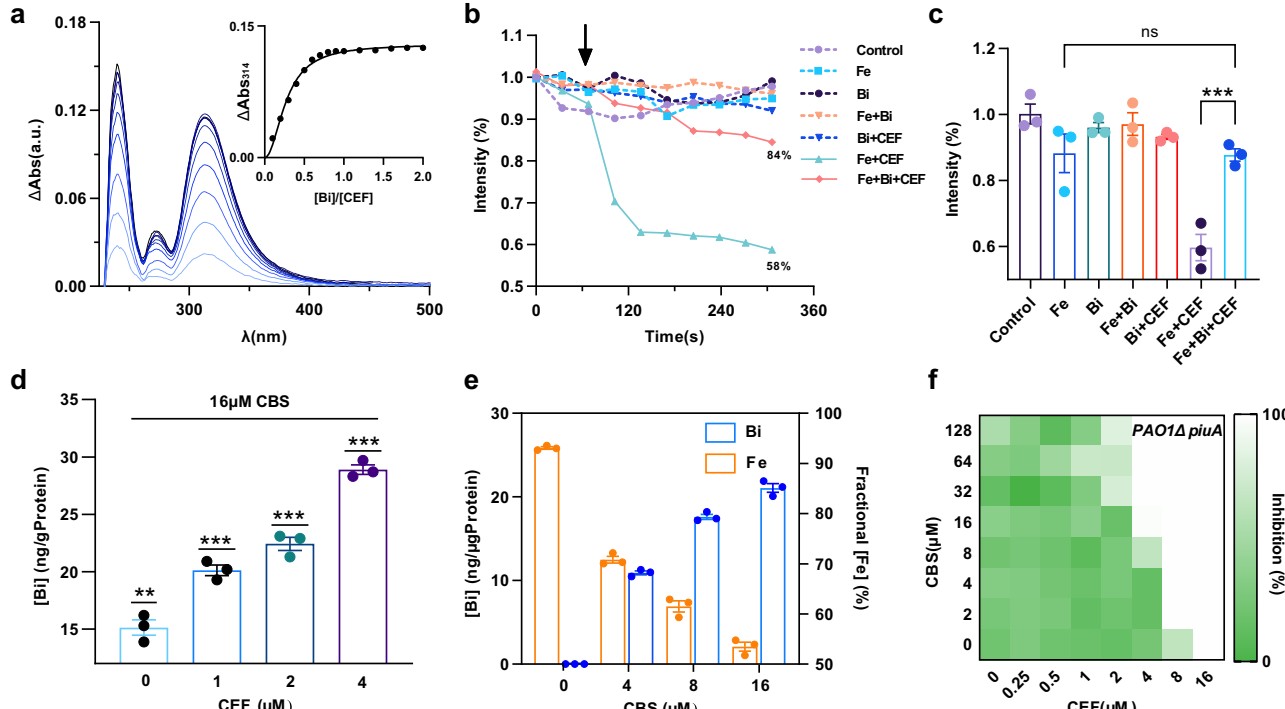

**Fig. 3 | Mechanistic insights into synergy between bismuth drugs and CEF.**
**a** Difference UV-vis spectra of CEF upon addition of 0.1–1.0 molar equivalents of CBS. A titration curve at 314 nm is shown as the insert. HEPES buffer at pH 7.4 was used. **b** Iron uptake by PAO1 detected by calcein-AM fluorescence intensity in the bacterial cells. During 300 s of monitoring, CEF at 50 μg ml$^{-1}$ was added at 60 s after the initiation of the monitoring. Calcein-AM coordinates with iron in bacterial cells, resulting in fluorescence to be quenched. The concentration of bismuth and iron compounds are 50 and 3 μM, respectively. **c** Final intensity of fluorescence in PAO1 after 1 h treatment. Mean value of three replicates are shown and error bars indicates ±SEM. $p$ values were determined by one-way ANOVA test. ***$p$ = 0.0002, ns $p$ > 0.99. **d** Bismuth concentration per protein in cell lysates in the presence of

different concentration of CEF determined by ICP-MS. CBS at 16 μM was supplemented to the bacterial cells for 30 min. $n$ = 3 biologically independent samples. Error bars indicates mean ± SEM. $p$ values were determined by two-tailed one sample $t$-test, **$p$ = 0.0019, ***$p$ < 0.001. **e** The uptake of bismuth and iron in bacterial cells treated with different concentration of CBS and 0.5 μM of CEF. Bismuth concentration was determined by ICP-MS and Fe concentration was determined by calcein-AM and normalized as percentage. $n$ = 3 biologically independent samples. Mean value of three replicates are shown and error bars indicates ±SEM. **f** Representative heat plots of microdilution checkboard assay for the combination of CEF and CBS against PAO1ΔPiuA. Abs absorbance, Fe iron citrate, Bi colloidal bismuth citrate, CEF cefiderocol. Source data are provided as a Source Data file.

suggesting the combination of CEF and CBS significantly suppressed the evolution of high-level resistance in PAO1.

To further assess whether the combination could be potentially used to treat infections caused by CEF-resistant *P. aeruginosa*, we first generated five CEF-resistant strains, including three PAO1 strains and two clinical isolates PA1882 by treatment of these bacteria with a sub-inhibitory concentration of CEF (started from 0.5 μM) over a period of 12 consecutive passages, and the MICs after 12 passages were measured to be 16 μM, which are higher than the breakpoint of CEF (8 μM) defined by Investigational CLSI MIC, confirming these bacteria strains are resistant to CEF[38]. We then determined the MICs of CEF against these resistant bacterial strains in the presence of various concentrations of CBS by the standard checkboard microdilution method. In the presence of 32 μM CBS, about 8-fold and 4-fold drops in the MICs of CEF were found for CEF-resistant clinical isolate PA1882-R1 and PAO1-R1 (Supplementary Fig. 8) respectively, compared with those without CBS, indicating that the combination of CEF with CBS could resensitize these resistant *P. aeruginosa* to CEF. Moreover, the fractional inhibitory concentration indexes (FICIs) were determined to be lower than 0.5 in all resistant strains, indicative of synergies between CBS and CEF against all CEF-resistant bacterial strains (Fig. 2e). We further evaluated the efficacy of the co-therapy against clinical CEF-resistant *P. aeruginosa* isolates PA247 and PA245. Interestingly, in the presence of 32 μM CBS, the MICs of CEF against both PA247 and PA245 were reduced from 16 to 1 μM, suggesting that CBS could resensitize these resistant bacterial strains to CEF (Fig. 2f, g), and strong synergies were observed

between CBS and CEF against these clinical isolates as judged by the FICIs of 0.125 and 0.188. Taken together, the combination of CEF and CBS can not only suppress the CEF resistance evolution of *P. aeruginosa*, but also potentially treat the infections caused by CEF-resistant strains, suggesting a potential clinical usage of the combination therapy to prolong the life-span of CEF to fight against antimicrobial resistance.

### Binding of Bi(III) to CEF results in enhanced uptake of Bi(III) and reduced levels of Fe(III) in PAO1

To obtain further mechanistic insight into the synergistic effect of CBS with CEF, we first examined whether Bi$^{3+}$ is bound to CEF by UV-vis spectroscopy, NMR spectroscopy and MS spectrometry. Titration of CBS to CEF, which was dissolved in a mixture of HEPES buffer and dimethyl sulfoxide (DMSO), resulted in the appearance and increase in the absorbance at 314 nm in the UV difference spectrum, indicative of binding of Bi$^{3+}$ to CEF. The absorption at 314 nm was plotted against CBS concentration (Fig. 3a), the molar ratio of Bi$^{3+}$ to CEF was found to be 0.7 (±0.2), and the $K_d$ was calculated to be 10.35 μM through the Ryan-Weber equation. The binding was also confirmed by $^1$H NMR spectroscopy (Supplementary Fig. 9a–c). The two peaks at 6.82 and 6.73 assignable to the aromatic protons decreased their intensities gradually and finally disappeared upon the addition of different molar equivalents of CBS, and new peaks appeared at 7.05 and 6.93 ppm initially and broadened upon addition of more CBS, indicating that Bi$^{3+}$ binds the chlorocatechol, and the free and bound CEF are in an

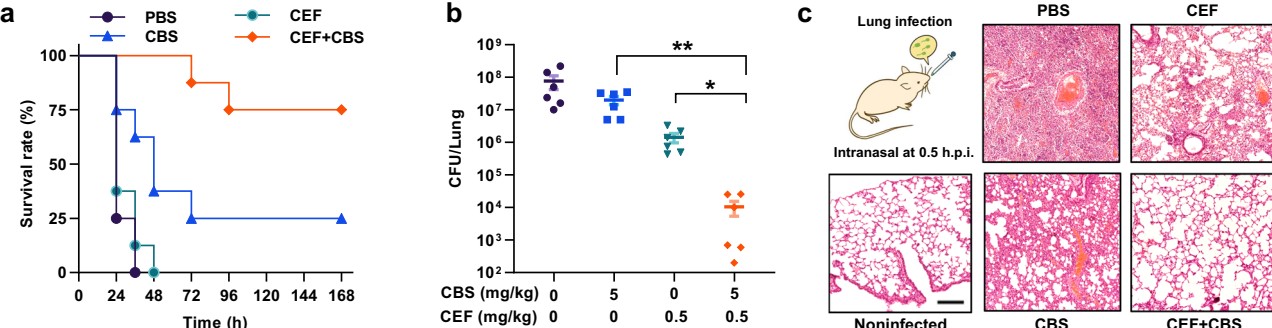

**Fig. 4 | CBS enhances the antibacterial activity of CEF against PAO1 in vivo.**
**a** Survival curves showing efficacies in a murine acute pneumonia model ($n = 8$).
BALB/c mice were infected by a lethal dose of *P. aeruginosa* via intranasal inoculation. Four groups of mice were treated with a PBS, monotherapy of CEF (0.25 mg/kg),
CBS (5 mg/kg), or combination therapy of CEF and CBS. Increased survival rates of
mice over 7 days challenging were seen in the combination group. **b** Bacterial load in
a mouse lung infection model. *p* values were determined by two-tailed unpaired

*t*-test. \**p* = 0.0118; \*\**p* = 0.0059, which means significant difference compared with
control or single treated group. $n = 6$ biologically independent animals. The data
were presented as means ± SEM. **c** Hematoxylin and eosin (H&E) staining of the
mouse lungs under the different treatment post infections. Scale bar: 50 μm.
Experiments were performed in triplicate. CEF cefiderocol, CBS colloidal bismuth
citrate. Source data are provided as a Source Data file.

exchange on the NMR time scale. The MS spectrometry was also performed to further validate the binding. CEF gave rise to a peak at *m/z* of
752.1599 (Cald. 752.1570) (Supplementary Fig. 10), and incubation of
bismuth nitrate with CEF led to new peaks appeared at *m/z* of 958.1157
assigned as (Bi-CEF + H)$^+$ (Cald. 958.1139) and 479.5638 assigned as (Bi-CEF + 2H)$^{2+}$ (Cald. 479.5605). Taken together, we demonstrate that Bi$^{3+}$
binds to CEF to form a 1:1 complex.

The binding of CBS to CEF might result in higher bismuth uptake,
which simultaneously reduces intracellular Fe$^{3+}$ contents. To test this,
the intracellular iron concentration in PAO1 was measured using the
calcein-acetoxymethyl (AM) fluorescence probe[19], which gave rise to a
green fluorescence when transported into bacterial cells quenched by
chelating with intracellular iron. The fluorescence of calcein-AM was
quenched immediately after the addition of CEF (0.5 μM) under iron-sufficient condition compared to the control group, indicating the
uptake of Fe$^{3+}$ was distinguishably improved in the presence of
external siderophore (Fig. 3b, c). However, addition of Bi$^{3+}$ (50 μM) to
the group of Fe + CEF resulted in much higher fluorescence intensity,
indicating less Fe$^{3+}$ was uptaken by siderophore owing to the competition by Bi$^{3+}$. Other groups, i.e., Fe + Bi, Bi + CEF and Bi also exhibited
comparable fluorescence intensity to both the negative control group
and Fe$^{3+}$ individual group, indicative of no significant changes of the
intracellular iron concentration in these groups owing to either less
iron available (Bi + CEF and Bi groups) or inefficient transport of iron
(Fe + Bi group). The final fluorescence intensity is also in agreement
with the observation that the addition of bismuth decreased intracellular iron concentration substantially. We further measured intracellular bismuth levels by ICP-MS. As shown in Fig. 3d, when PAO1 was
treated with 16 μM CBS, the addition of increasing concentration of
CEF resulted in increased intracellular Bi$^{3+}$ concentrations, suggesting
more bismuth is transported by CEF into bacterial cells through
siderophore-metal uptake pathways. Upon the treatment of PAO1 with
different concentrations of CBS ranging from 0–16 μM, the Fe$^{3+}$ level in
the bacterial cells decreased accompanied by the increase in the Bi$^{3+}$
concentration (Fig. 3e). To further investigate the molecular mechanism of this synergistic effect, we constructed a PAO1 mutant with *piuA*
gene deletion (PAO1Δ*piuA*). The protein piuA was considered as one of
the iron transporters of *P. aeruginosa*, responsible for the transportation of CEF-iron complex[20,39], and the deletion of the *piuA* gene resulted in a higher MIC (16 μM) of CEF against PAO1 compared with that for
the wide-type PAO1 (2 μM). Moreover, only a 4-fold decrease in the MIC
of CEF against PAO1Δ*piuA* was observed (Fig. 3f) after co-treatment
with CBS up to 128 μM, and the FICI was calculated to be 0.5, indicating
two drugs are additive and independent without any synergy. These

results demonstrate that the synergy between CEF and CBS is abolished by the deletion of the CEF-metal complex transporter, and the Bi-CEF complex may partially be transported into the bacterial cells
through one of the CEF transporters piuA. Moreover, compared with
the untreated cells, PAO1 cells were elongated upon the treatment,
indicating that the bacterium undergoes anaerobic respiration[40] owing
to the iron-deficiency induced by Bi$^{3+}$. The membrane integrity was
also disrupted, which may increase the membrane permeability of CEF
(Supplementary Fig. 11). Collectively, we demonstrate that Bi$^{3+}$ could
coordinate with CEF via its chlorocatechol and compete with Fe$^{3+}$ for
cellular transport via the siderophore-mediated pathway.

### The combination of CBS and CEF shows high potency in vivo
We next first examined the potential toxicity of the combination of
CBS and CEF in mammalian cells using the human lung adenocarcinoma cell A549 and embryonic kidney cell HEK293T as examples by
XTT assay. With the increasing concentration of CEF combined with
100 μM CBS, there is no significant difference in cell viability between
negative control and the treated cells (Supplementary Fig. 12a, b).
Similarly, the cells treated with CBS and CEF combination at
each concentration up to 128 μM exhibited high cell viability >78%
(Supplementary Fig. 12c), demonstrating the low cytotoxicity of
CBS-CEF combination under effective inhibitory concentration
against PAO1.

Finally, we evaluated the efficiency of the CBS and CEF combination in vivo. Given that *P. aeruginosa* is one of the most common
pathogens causing respiratory infections in hospitalized patients[41], we
selected a murine acute pneumonia model for the study. Female 6- to
8-week BALB/c mice were infected via an intranasal inoculation of a
lethal dose ($6 \times 10^7$ CFU) of *P. aeruginosa* PAO1. The infected mice were
then administrated intranasally with a PBS, CEF or CBS monotherapy
or combination therapy half-hour post-infection. As shown in Fig. 4a, a
single dose of the combination of CBS (5 mg/kg) with CEF (0.25 mg/kg)
significantly increased the survival rates of mice to 75% compared to
CBS or CEF alone treated groups. In consistent with this, in a separated
experiment, in which mice were infected with a sub-lethal dose of *P.
aeruginosa* PAO1 ($4 \times 10^7$ CFU), the bacterial loads in the lungs of
infected mice showed evident decreases in the group treated with the
combination of CBS (5 mg/kg) and CEF (0.25 mg/kg) (Fig. 4b) but not in
the groups receiving monotherapies. Furthermore, the histological
sections of the lungs also demonstrated the recovery of the tissue
injury after the co-therapy of CEF and CBS (Fig. 4c). Taken together, we
demonstrate the in vitro efficacy of the CEF-CBS combination therapy
could be well translated into in vivo efficacy.

## Discussion

The emergence and spread of antimicrobial resistance in Gram-negative bacteria are on-going global threats, resulting in limited therapeutic options for severe infections. There is an urgent need for new antibiotics or new strategies to restore the efficacy of clinically used antibiotics. Inspired by the discovery of natural sideromycins, albomycin and salmycin, against Gram-negative bacterial infections[9], scientists have designed and synthesized vast libraries of side-romycins, using a Trojan Horse strategy to deliver antibiotics into bacterial cells via siderophore-mediated pathways[16]. Amongst all sideromycins, a chlorocatechol-ceftazidime sideromycin named cefiderocol exhibited potent activities against *P. aeruginosa* and other Gram-negative bacteria, and was approved for clinical use in 2019 in the US to treat complicated urinary tract infections[42]. However, the acquisition of CEF resistance has been reported very soon[21,43]. A case study showed that the in vivo development of CEF resistance occurred within 3 weeks after the therapy was initiated in a critically ill-patient with bloodstream and intro-abdominal infection caused by carbapenem-resistance *Enterobacter*, possibly due to mutation in the *cirA* gene encoding a catecholate siderophore receptor[44]. Thus, novel strategies are urgently needed to extend the life-span of CEF, or other sideromycins.

In this study, we show that Bi(III) drugs and Ga(III) drugs could enhance the efficacy of CEF against *P. aeruginosa* and *B. cepacia* strains as evidenced by a 64-fold reduction in the MIC of CEF. Using CBS as a showcase, we further demonstrated the strong synergy between CBS and CEF by a low FICI value (i.e., 0.125) and time kill assay (Fig. 1d). The in vitro interaction of $Bi^{3+}$ with CEF was confirmed by both UV-vis spectroscopy and MS spectrometry, which showed the formation of a 1:1 complex of $Bi^{3+}$-CEF (Fig. 3a). With the increase in bismuth(III) concentration, more $Bi^{3+}$ might bind to CEF and is likely to be transported to cell through competition with $Fe^{3+}$, resulting in a much lower level of $Fe^{3+}$ (Fig. 3b). This is further validated by our PAO1Δ*piuA* data, suggesting PiuA plays a role in the transport of both Bi-CEF and Fe-CEF. Given that $Bi^{3+}$ was previously demonstrated to exhibit antimicrobial activity through inhibiting multiple enzymes in *H. pylori*[45,46], it is highly possible that enhanced uptake of bismuth might also bind and functionally disrupt multiple key enzymes in *P. aeruginosa*. Significantly, we show that the combined use of CBS and CEF could not only efficiently suppress the development of high-level bacterial resistant mutant to CEF, but also inhibit the growth of CEF-resistant *P. aeruginosa* including clinical isolates. This is probably attributable either to the reduced dosage of CEF in the combination therapy, or the suppressed levels of $Fe^{3+}$ in bacterial cells caused by the competition by $Bi^{3+}$, which may render the bacterium unlikely to shut down the iron transport pathways including the siderophore pathways. Further studies are warranted in future to validate this. Significantly, the combination of a bismuth drug and CEF exhibits low toxicity to mammalian cells and high potency in a murine lung infection model compared with monotherapy (Fig. 4a, b), confirming that such a co-therapy may readily be translated into clinical usage.

In summary, we demonstrate a bismuth drug (CBS) could enhance the potency of CEF both in vitro and in vivo against *P. aeruginosa*, suppress the development of high-level bacterial resistance to CEF, and restore the efficacy of CEF against resistant *P. aeruginosa* including clinical isolates. Such phenomena are likely attributable to the competition of $Bi^{3+}$ with $Fe^{3+}$ to CEF, leading to the reduced uptake of $Fe^{3+}$ and enhanced uptake of antimicrobial $Bi^{3+}$/$Ga^{3+}$ as well as disruption of bacterial membrane integrity, thus increasing antibiotic permeability. The metallo-sideromycin might not only improve the efficiency of sideromycin, but also prolong the effective life-span of this type of antibiotics. It is worth of further investigation of other sideromycins and metals, to thoroughly explore the potentials of metallo-sideromycins in the treatment of infections caused by drug-resistant bacterial pathogens.

## Methods

The research complies with all relevant ethical regulations. All animal experiments were approved by and performed in accordance with the guidelines approved by Committee on the Use of Live Animals in Teaching and Research (CULATR), the University of Hong Kong.

### Chemicals

Cefiderocol (CEF) was purchased from ChemScene LLC (USA). Mueller–Hinton Broth (MHB), Luria-Bertani (LB) Broth Powder, CA-MHB Broth Power, M9 Broth Powder and Agar powder were purchased from Affymetrix. Tryptone and Yeast extract were purchased from Oxoid Ltd. (UK). dNTPs, pfu DNA polymerase, Taq DNA polymerase, and DNA marker were purchased from Thermo Fisher Scientific Ltd. Restriction endonucleases (BamH I, Hind III, Sac I) were purchased from TAKARA. Plasmid-Mini-prep Kit and Genomic DNA Purification Kit were purchased from Thermo Fisher Scientific Ltd. All other chemicals were purchased from Sigma-Aldrich unless specified.

### Bacterial strains and growth conditions

The strains, plasmids and primers used in this study are listed in Supplementary Table 3. Primers were purchased from BGI. Bacteria strains were cultured in lysogeny broth (LB) at 37 °C shaken at 250 rpm if not mentioned. *E. coli* DH5α and S17 were used as the cloning host and the donor bacterial for biparental matings respectively. Anti-biotics were used at the following concentrations: for *E. coli*, kana-mycin 50 μg/ml, tetracycline 10 μg/ml; for *P. aeruginosa*, tetracycline 50 μg/ml, carbenicillin 150 μg/ml. Sucrose-resistant colonies were selected by sucrose-agar plate, which was supplemented with 5 g/l yeast extract, 10 g/l tryptone, 75 g/l sucrose, 1.5% (w/v) agar powder and deionized water.

### Construction of knockout mutants

The construction of unmarked knockout mutants was based on the method of Tung T Hoang et al. with slight change[47]. Briefly, the primer A1/A2 and B1/B2 were designed from the PAO1 genome sequence as a template, and the DNA fragments of ~1000 bp upstream and down-stream of the target knockout gene were PCR-amplified based on A1/A2 and B1/B2 primers. The fragments were purified and cloned into the plasmid pEX18TC, followed by transferring into the *E. coli* S17 and then into *P. aeruginosa* PAO1 through biparental mating. Single crossover mutants were selected by 50 μg/ml tetracycline and 50 μg/ml kana-mycin to kill the *E. coli* donor strains. Double crossover mutants were selected by sucrose-agar plates.

### Microdilution MIC assay

MIC values were assessed by the standard broth microdilution assay in accordance with Clinical and Laboratory Standards Institute (CLSI) guidelines[48]. Bacterial cells were grown overnight on LB agar plates to obtain single colonies. Three morphologically distinct colonies were transferred to CA-MHB broth and cultured overnight at 37 °C shaken at 250 rpm. Then the $OD_{600}$ was measured to adjust the bacterial density to about $1 \times 10^6$ CFU ml$^{-1}$ and checked by CFU counting on agar plates. The flat-bottomed 96-well plates were prepared, and CEF/bismuth or other metal compounds were added into the first column of 96-well plates and given 2-fold dilution, followed by adding bacterial inocula prepared before. The total volume should be within 100 μl per well. The plate was then incubated at 37 °C overnight. The group without antibiotic/bismuth compounds served as growth control and the group without bacterial cells served as background control. The OD was measured at 600 nm with Multi-Mode Microplate Readers (Spec-traMax iD3, Molecular Devices, LLC.), and the MIC was read as the lowest concentration of bacterial cells that had no visible growth and resulted in an $OD_{600} < 0.04$ after normalization to the untreated growth control and the background control.

For the checkboard microdilution assay used for drug combination test, cefiderocol and bismuth compounds were both 2-fold diluted from two-dimensional orientation. The FICI was calculated using the following equation:

$$FICI = \frac{C_{cef,combination}}{C_{cef,alone}} + \frac{C_{metal\ compounds,combination}}{C_{metal\ compounds,alone}} \quad (1)$$

Synergism was defined as FICI < 0.5 and antagonism as FICI > 4. All experiments were replicated in triplicate.

## Time kill assay

To further illustrate the synergistic effect between CEF and CBS, time kill assay was performed with slight change[29]. PAO1 was cultured overnight on LB broth at 37 °C. Then the bacterial cells were 1:250 diluted with LB broth and cultured for 3 h at 37 °C to reach log phase, followed by adjusting the bacterial density to about $1 \times 10^6$ CFU ml$^{-1}$ using an $OD_{600}$ reader. Then the bacterial cells were exposed to CEF, CBS alone or their combination. The concentrations of CBS are 16 and 4 μM, and the concentrations of CEF are 1 and 0.5 μM respectively. No drug group served as a positive control. Ten μl of the suspension were aspirated from each group at different time interval from 0 to 24 h (0, 1, 2, 4, 6, 12, 24 h) for inspection of bacterial viability by agar plating. All experiments were performed in triplicate.

## Resistance study

For the MPC study[49], PAO1 was plated on LB agar plate with CEF and CBS at different concentrations and incubated at 37 °C. Then more than four colonies were picked from any plates with colonies and re-cultured to measure the MIC values. Any colony with higher MIC than the original values was determined as a higher-level resistant mutant colony. The measured concentrations that could inhibit the growth of mutant colonies was determined as MPC. For the serial passage assay, the PAO1 strains were cultured overnight and diluted to ~$10^7$ CFU ml$^{-1}$ in LB broth. The bacterial suspension was added to a 96-well plate with 2-fold dilution of drugs starting from 4-folds, and 32 folds of MIC after 16 passages. All the plates were incubated at 37 °C and the growth of cultures was checked at 24 h intervals. Cultures from the second highest concentrations that allowed growth were diluted 1:1000 into fresh medium, following addition of same concentrations of drugs. This serial passaging was repeated daily for 20 days. The MICs of CEF in the presence of CBS were determined for every passage. At the end-point of the evolution, single colonies from each group were selected and stored in 15% glycerol at −80 °C.

## Biofilm formation measurement

Biofilm formation was firstly measured by Crystal violet (CV) method. PAO1 was used as the tested strain. Three distinct colonies from LB agar plate were transferred to LB medium and cultured at 37 °C shaken at 220 rpm overnight. The overnight culture was then 1:100 diluted with CA-MHB medium and 100 μl of the dilution were added per well in a 96-well plate and exposed to CEF, CBS either alone or combination. Untreated group served as positive control. The microtiter plate was incubated for 48 h at 37 °C. After incubation, biofilm was washed gently with PBS and water three times to remove unattached cells, followed by adding 150 μl of a 0.1% crystal violet solution. Then the plate was incubated at room temperature for 15 min to stain the biofilm. After incubation, each well was washed by water 3 times to completely remove the dye and turn the plate upside down to dry overnight at room temperature. To quantify the biofilm formation, 150 μl 30% acetic acid solution were used for each well to dissolve the crystal violet. After incubation for 10–15 min at room temperature, 100 μl of each well were transferred to a new flat-bottomed microplate for $OD_{550}$ reading by Multi-Mode Microplate Readers (SpectraMax iD3, Molecular Devices, LLC.). In total, 30% acetic acid in water served as a blank. Four

replicated wells for each group were used. For the Confocal imaging, PAO1-GFP strain with carbenicillin resistance was used for fluorescent monitoring. PAO1-GFP bacterial cells were grown overnight on LB agar plate with 150 μM carbenicillin to obtain single colonies. Three distinct colonies were transferred to LB medium with 150 μM carbenicillin and cultured at 37 °C shaken at 220 rpm overnight. Then the bacterial cells were 1:100 diluted with fresh LB medium in glass bottomed confocal plate. CBS and CEF were added individually or combinedly as different treated groups in triplicate and LB broth with no drugs served as a control group. After 48 h incubation at 37 °C, the supernatant was removed and the biofilm was washed with PBS three times for removing the suspended cells. Then fresh LB medium was added into each well for directly imaging by confocal laser scanning microscopy (Leica TCS SP8 CLSM). Fluorescein was excited at 488 nm.

## Cellular metal uptake monitored by calcein-AM

Briefly, PAO1 was cultured overnight in M9 iron-deficient medium, collected by centrifugation, and resuspended to adjust $OD_{600}$ to 5 in M9 medium. Calcein-AM was added at a final concentration of 20 μM to incubate for 120 min at room temperature under light-shielded conditions. Chloramphenicol at a concentration of 97.9 μg/ml, at which the bacterium would not be killed, was also added at the same time to suppress the cell activity. The cell was then washed by PBS buffer and resuspended to adjust $OD_{600}$ to 5 in M9 medium following transferred to a 96-well plate. The fluorescence intensity was monitored by a Multi-Mode Microplate Readers (SpectraMax iD3, Molecular Devices, LLC.) with 492-nm excitation and 535-nm emission. Then $Fe^{3+}$ (as iron citrate) and $Bi^{3+}$ (as CBS) were added at a final concentration of 3 and 50 μM respectively to give rise to three groups, i.e., iron (Fe) alone, bismuth (Bi) alone and Fe+Bi. CEF was added at 60 s after the initiation of the monitoring at a final concentration of 0.5 μM.

## Metal uptake measured by ICP-MS

Three colonies were transferred to LB broth and cultured overnight at 37 °C shaken at 220 rpm. The bacterial cells were then collected by centrifugation and resuspended to adjust the $OD_{600}$ to 2.5 in LB medium. CEF at different concentrations (0, 1, 2 and 4 μM) was added to the respective groups in triplicate followed by addition of 16 μM of CBS. LB broth with no drugs served as a control group. After 30 min incubation, bacterial suspensions for each group were collected and washed with PBS buffer for three times, followed by the resuspending by Tris buffer (pH 7) in the presence of a protease inhibitor cocktail (PIC) solution (1%) and then sonicated on ice. All the samples were centrifuged 20 min at $12,470 \times g$ at 4 °C to remove the cell debris after sonication, then 200 μl of cell lysate from each group were taken to dissolve to 200 μl of 68% $HNO_3$ at 60 °C overnight. The dissolved sample was diluted to appropriate concentration for ICP-MS (Agilent 7500a, Agilent Technologies, CA, USA) by 1% $HNO_3$. $^{115}$In served as an internal standard. Protein concentration of cell lysate was quantified by BCA quantification kit.

## UV-vis spectroscopy

UV-vis spectra were recorded on a Varian Cary 50 spectrophotometer at a rate of 360 nm/min using a 1-cm quartz cuvette. CEF stock solution was diluted by 10 mM HEPES at pH 7.4 to a final concentration of 20 μM. CBS was stepwise titrated into cefiderocol solution and UV-vis spectra were recorded in a range of 200–800 nm. HEPES buffer (10 mM) was measured as baseline correction. CBS was titrated with a 2 μM interval from 0 to 40 μM, and the binding of bismuth to CEF was monitored by the increase in absorption at 340 nm. Difference spectra were obtained by subtracting the spectra of CEF. The UV-vis titration curve was fitted to Ryan-Weber nonlinear equation[50]:

$$I = \frac{I_{max}}{2C_p}[(K_d + C_m + C_p) - \sqrt{(C_p + C_m + K_d)^2 - 4C_mC_p}] \quad (2)$$

where $I$ represents UV-vis intensity; $I_{max}$ is the maximum UV-vis intensity; $K_d$ is the dissociation constant; $C_p$ is the total concentration of CEF and $C_m$ is the concentration of CBS.

## SEM imaging for bacterial cells

Three colonies were transferred to LB broth and cultured overnight at 37 °C shaken at 220 rpm. Then the bacterial cells were collected by centrifugation ($1494 \times g$, 10 min) and resuspended to LB medium to adjust OD600 to 1. CEF and CBS were added individually or combinedly at the concentration of 0.5 and 16 µM respectively. After 30 min incubation, bacterial suspensions for each group were collected and washed with PBS buffer three times, followed by adding 2.5% glutaraldehyde overnight at 4 °C to fix bacterial cells. The fixed cells were washed by PBS three times to remove glutaraldehyde. The sample dehydration was performed by different ethanol/water volumes ranging from 30, 50, 70, 80, 90, and 100%, and incubated for 10 min for each ethanol volume. Then the ethanol was dried at room temperature. After coating with gold, SEM imaging was performed by High Resolution Schottky FE-SEM, TESCAN MAIA3 XMN, and images were collected by Maia TC software.

## NMR study

NMR experiment was performed on a Bruker 500 MHz NMR spectrometer. CEF was dissolved in a mixed solvent of deuterated dimethyl sulfoxide (DMSO-$d_6$) and water ($H_2O$) at a final concentration of 1 mM. CBS was titrated into CEF solution stepwise as a 0.2 molar equiv interval and the $^1$H NMR was recorded after reaction for at least 5 min. Binding of bismuth ions ($Bi^{3+}$) to CEF was evidenced by the appearance of the resonances at 7.15 ppm and 6.93 ppm.

## Mass spectrometry (MS)

MS samples were prepared by mixing CEF and bismuth nitrate in water, following by addition of $NH_4OH$ to adjust the pH to ~7. The mixture was incubated at room temperature for 1 h with gentle shaking. Then the mixed solution was diluted by methanol to a final concentration of 1 µM for the MS examination. Sodium formate was used as an internal standard. Mass Spectrometry was performed and analyzed by UHR TOF LC-MS system, Bruker Maxis II and Bruker data analysis.

## Cytotoxicity study

The cytotoxicity of CEF combined with CBS was evaluated by MTT assay in human lung adenocarcinoma Cell A549 and HEK-293T Cell in Dulbecco's Modified Eagle Medium (DMEM, with 10% FBS 1% p/s). A549 (86012804) and HEK-293T (12022001) cells were purchased from Sigma-Aldrich. Cells were seeded in a 96-well culture plates at ~3000 cells/well 1 day before compound addition. CEF was firstly performed by serial dilution using DMSO in a DMSO-resistant 96-well plate. A 96-well plate with 300 µl DMEM containing 1% FBS per well was prepared and 3 µl of serial-diluted CEF were added following by CBS. DMEM with no drugs served as a negative control. After washing the cells twice with PBS, 100 µl of compound-containing medium were added into the cells to get a final concentration of 100, 50, 25, 12.5, 6.25, 3.125, 1.56, 0.78, 0.39 and 0.2 µM for CEF, and 100 µM per well for CBS. The cells were incubated with the compounds for 24 h at 37 °C under $CO_2$ protection. After overnight incubation, 10 µl of MTT solutions (5 mg/ml) were added into each well and incubated for 4 h, then 100 µl 1% SDS in 0.01 M HCl per well were added and incubated overnight. The optical density (OD) was measured at 570/640 nm with Multi-Mode Microplate Readers (SpectraMax iD3, Molecular Devices, LLC.). Cell viability was calculated using the following equation:

$$\text{Cell Viability}(\%) = 100\% \times \frac{E_{sample} - E_{blank}}{E_{control} - E_{blank}} \tag{3}$$

## Animal study

All animal experiments were approved by and performed in accordance with the guidelines approved by Committee on the Use of Live Animals in Teaching and Research (CULATR), the University of Hong Kong. A total of 6–8 weeks-old, female BALB/c mice were used in all mouse studies. Mice rooms have 12-h dark-light cycle, the room temperature is around 24 °C, and the humidity is kept at 45–65%. Murine acute pneumonia model was performed in this study. No sex-based analyses have performed in this study. Briefly, PAO1 were cultured overnight at 37 °C shaken at 220 rpm, following by 1:100 dilution in fresh LB and cultured at 37 °C to OD$_{600}$ of 1. The bacterial cells were collected and adjusted the concentration to $3 \times 10^9$ CFU ml$^{-1}$ by PBS after washing. Each 6- to 8-week-old female BALB/c mouse was anesthetized by an intraperitoneal injection of ketamine (100 mg/kg) and xylazine (10 mg/kg), followed by an intranasal inoculation of 20 µl of the bacterial suspension. After half-hour post-infection, the mice were administered with 20 µl aliquot of PBS, CBS, CEF, or their combination intranasally. Body weights and mouse survivals were monitored till endpoint of the experiment. For the lung colonization and histological study, each 6- to 8-week-old female BALB/c mouse was take an intranasal inoculation of 20 µl of the bacterial suspension, resulting in $4 \times 10^6$ CFU ml$^{-1}$ per mouse. After 24 h post treatment, the mice were euthanized by intraperitoneal injection of 100 mg/kg pentobarbitone. The bacterial loads were counted by agar plating. For the histological study, lung samples were stored in 10% formalin for 48 h and rinsed by 70% EtOH. Tissues were embedded in paraffin, thin-sectioned, stained with hematoxylin and eosin (H&E) and examined by microscopy.

## Statistical analysis

All statistical analyses were performed on three independent experiments, or more if otherwise stated, using Prism 9.0 (GraphPad Software Inc.) software. NMR data were analyzed by MestReNova X64 14.3.3.

## Reporting summary

Further information on research design is available in the Nature Portfolio Reporting Summary linked to this article.

## Data availability

All data generated in this study are provided in the Supplementary Information and Source data file. Source data are provided with this paper.

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

## Acknowledgements

We thank Mr. Xu Wei for the help in SEM imaging, Ms. Jo for the help in MS spectrometry, Dr. Gao Peng for the help in animal studies. This research was supported by the Research Grants Council of Hong Kong SAR (R7070-18, 17308921, 2122-7S04), the Health and Medical Research Fund of the Health Bureau of Hong Kong SAR (CID HKU1-13) and the University of Hong Kong (URC (202107185074) and Norman & Cecilia Yip Foundation).

## Author contributions

H.S., H.L. and R.W. conceived idea and designed the project; C.W. performed the antimicrobial experiments and data analysis; Y.X. performed animal experiments and the screening of clinical isolates; R.Y.T.K. and P.L.H. performed the collections of clinical isolates; J.L. provided the eukaryotic cell lines on cell toxicity experiments; C.W. performed the characterization of metal complex. Y.X., C.L.C. and P.H.T. provided suggestions; C.W., H.L. and H.S. prepared the manuscript with input from all.

## Competing interests

The authors declare no competing interests.
