## [Peer Review File · Nature Communications]

REVIEWER COMMENTS

Reviewer #1 (Remarks to the Author):

While the concept of binding non-iron metals to siderophores and sideromycins to generate novel antibiotics is not new, this detailed study of cefiderocol metal (notably bismuth) is noteworthy. As noted, cefiderocol is the first marketed antibiotic containing an iron chelating group (chloro dihydroxy benzoate) that enables it to use siderophore mediated active transport processes to improve targeted delivery and activity against primarily Gram-negative bacteria.

Studies of sideromycins (siderophore-antibiotic conjugates) have revealed the tremendous potential of this so called "Trojan Horse" antibiotic approach. However, as most papers related to cefiderocol improperly state, the iron binding component of cefiderocol is NOT a siderophore but a mimic of the natural bis catechol components of real natural siderophores. Thus, throughout the paper, this should be clarified. Although now a bit older, still the best overall reference that should be included early in this paper (not later as reference 36) is that by Hider (Hider, R. C.; Kong, X. Chemistry and biology of siderophores. Nat. Prod. Rep. 2010, 27, 637–657). The appendices associated with that review provide wonderful sources of the structures of siderophores, their microbial producers and a wealth of additional information that is essential reading for anyone in the area.

Some background is given to the sideromycin Trojan Horse strategy, but as also with most related papers, key references are missing. Over the last couple of decades, controversy has impeded the development of the potential of sideromycins (see reference 12 in the manuscript); however, this can unfortunately be attributed to the lack of reference to much earlier papers that demonstrate the clinical potential of natural sideromycins like albomycins. The authors reference albomycins via the Hider review (now ref 36), but do not include the essential references that demonstrate the efficacy in mammals, including thousands of patients. It is imperative that G. Benz, et al be given full credit for the first synthesis (which also determined the real structure) in the 1980s. Benz, G.; Schroeder, T.; Kurz, J.; Wuensche, C.; Karl, W.; Steffens, G.; Pfitzner, J. Konstitution der Desferriform der Albomycine δ_1 , δ_2 , ϵ . 1982, , 1322–1335. This pioneering and most impressive work must be referenced. The authors must include a reference to the extensive efforts by V. Braun and group related to studies of the albomycins and salmycins but should indicate that Braun demonstrated in vivo efficacy in animal studies: Biometals (2009) 22:3–13 DOI 10.1007/s10534-008-9199-7. Most importantly, the authors must include in the albomycin discussion the early studies by Gause on the discovery of albomycins and especially the report that they were used successfully in thousands of people in the Soviet Union in the late 1940s-50s and were remarkably effective in humans infected with already penicillin resistant bacteria. This work is seldom referenced, but attention to it would have avoided much of the controversy about the potential of sideromycins. This reference must be added along with related discussion: Gause GF (1955) Recent studies on albomycin, a new antibiotic. British Medical Journal 1776-1179.

Aside from the points made above, additional comments are provided below:

The authors state that they selected cefiderocol to showcase a series of metallo-compounds against different bacterial strains, but this sounds overly broad when the focus is on gallium and primarily

bismuth complexes of cefiderocol (though they did initially screen other metal complexes). The section on results provides good detail, including proper choice of media, (including iron sufficient and iron depleted media). They screened combinations of cefiderocol and bismuth and gallium salts against about 10 different classes of Gram-negative bacteria and only found an influence (32-64 reduction of MIC relative to cefiderocol itself) against the initial strains of *P. aeruginosa* and *B. cepacia*. The authors suggest that this selectivity might be due to the different bacterial strains having different iron requirements and uptake pathways, but this is curious since the other Gram-negative bacteria utilize catechol type siderophores. A better explanation is needed rather than the generalized speculation. With that initial determination of selectivity, the authors focused on *P. aeruginosa* strains and did proper controls by performing assays with cefiderocol alone, bismuth citrate alone and combinations in both iron sufficient and deficient media. The studies suggested that the combination of cefiderocol and bismuth gave improved activity to demonstrate their proposed “dual Trojan Horse” strategy.

Importantly, they also found that combination of cefiderocol and bismuth synergistically inhibited biofilm formation and suppressed resistance to cefiderocol in select strains of *P. aeruginosa*. Studies of isolated cefiderocol resistant strains indicated that combination of cefiderocol with bismuth resensitized the resistant strains to cefiderocol.

An additional important study was determination that bismuth binds to cefiderocol with 1:1 stoichiometry, consistent with the rather interesting and surprising claim by Shionogi that cefiderocol binds iron with 1:1 stoichiometry even though it contains only a single catechol unlike most siderophores or sideromycins. They also hypothesized that the bismuth complex would also suppress iron uptake and demonstrated consistent results using a relevant fluorescent probe. Also consistent with the active transport was the construction and study with a PAO1 mutant with a *piuA* gene (iron receptor) deletion.

Significantly, the authors demonstrated that the cefiderocol-bismuth complex was non toxic and somewhat effective in an *in vivo* study (an increase in survival rates to 75% compared to cefiderocol or bismuth citrate alone).

Overall, most of the results are consistent with the hypothesis that combination of cefiderocol and bismuth improves activity but only of select strains (which again is curious, since many of the bacteria studied and listed in Table 1, utilize catechols for iron transport). Further discussion of this selectivity is needed. Particularly, in most of the other strains in Table 1, the activity of the combination is essentially the same as for cefiderocol alone. It would be important to determine if in those cases, the bismuth conjugate was actually generated – the activity is consistent with the free non bismuth bound cefiderocol – why would it not form the bismuth complex in these cases? Also in Table 1, the activity of the other metal complexes is essentially the same as that for cefiderocol but it is well known that Chromium forms kinetically inert complexes with siderophores and, if cefiderocol also forms a chromium complex, it should have been recognized by the iron receptors and transporters and have been active like the bismuth complex. These results and the notable selectivity for *P. aeruginosa* need further study and elucidation.

If the changes suggested above are incorporated and IF the authors can provide data to rationalize the comments in the paragraph above, then the paper could be accepted in this high level journal. If not, I would recommend preliminary publication elsewhere.

The manuscript should be screened for minor grammatical typographical errors that occur through the paper and distract from the science. It is not the role of a scientific reviewer to make such corrections and should be done by authors and the editorial office.

Reviewer #2 (Remarks to the Author):

Sun and coworkers describe an investigation into the potential synergy of the sideromycin cefiderocol, a clinically approved antibiotic with metallodrugs/metal ions. The authors found that, sources of bismuth such as colloidal bismuth citrate see to effectively synergise with Cefiderocol in the treatment of *Pseudomonas aeruginosa*. This also translated somewhat to biofilm inhibition albeit without complete eradication. The binding of Bi(III) to cefiderocol was studied by standard techniques and it was shown that bacterial concentrations of bismuth increase in the presence of the sideromycin while iron concentrations decrease. Lastly the authors show that the cefiderocol-bismuth combination shows low toxicity in vitro and does show efficacy in an in vivo *P. aeruginosa* infection in mice. In this lung infection model the metal-sideromycin combination increased the survival rate and significantly reduced the bacterial load.

Overall this article by Sun and coworkers paints a promising picture for the usage of metal ions in conjunction with clinically approved antibiotics. The fact that both cefiderocol and CSB have been shown to be safe in humans would greatly accelerate the clinical development of such a combination. The one point of contention could be that this synergy seems to only apply to a fraction of the clinical *P. aeruginosa* population, so the applicability of this approach will have to be explored carefully. In the end I think this work presents noteworthy results worthy of publication in Nature Communications. There is an urgent need for new antibiotic strategies, particularly for Gram-negative ESKAPE pathogens, hence this work is of high significance. The conclusions are supported by the experimental data provided.

There are a few comments that should be addressed before publication:

- The authors state that the combination of cefiderocol with CSB was tested against 60 clinical strains of *P. aeruginosa* but was only found to be synergistic in 40% of the cases which might reduce the clinical relevance of this treatment. Do the authors know why the synergy was not effective in 60% of these strains?
- Against Biofilms the efficacy was found to be at maximum 80%. This seems suboptimal as I would assume a biofilm could simply regrow after treatment in such a scenario (was this observed)
- The NMR spectra given for the characterisation of the cefiderocol and bismuth are not ideal. Only a narrow area of the spectra is shown. The full NMR should be provided. Upon coordination of bismuth,

one would expect a shift also in the protons belonging to the aromatic region of the catechol. Also, the authors use deuterated protonated solvents (D₂O), this means the catechol protons could simply be exchanging with deuterium and hence disappear.

- In the in vivo infection model cefiderocol alone seemed to not be effective at all in increasing survival, why is that? Should it not be effective in principle?

Reviewer #3 (Remarks to the Author):

This is an important manuscript that explores synergistic effects between the Trojan horse antimicrobial cefiderocol and Fe(III) mimics, in particular Bi(III). Interestingly, bismuth citrate was found to enhance the potency of cefiderocol against certain *P. aeruginosa* strains, both in vitro and in vivo. It is highly significant that the combination was found to impede biofilm formation and delay the development of resistance. The authors demonstrate that Bi(III) competes with Fe(III) by coordinating to the catechol-part of cefiderocol and uptake studies suggest that the resulting Bi(III) complex is taken up via the iron-siderophore pathway. This is a potentially transformative development. Considering the increasing threat of antimicrobial resistance, the manuscript merits publication in Nature Communications.

There are, however, a number of points to be addressed before publication can be considered:

General points:

It was found that both Bi(III) and Ga(III) are acting synergistically with cefiderocol, however, only Bi(III) is mentioned in the abstract and in the conclusion section. Similarly, only colloidal bismuth citrate (CBS) was progressed to biofilm formation, resistance evolution, cellular uptake and in vivo studies. The authors should explain in the text why Ga(III) and the other Bi(III) salts that showed promise in the initial activity screens were not investigated further.

Synergism was also observed with certain *B. cepacia* strains. However, this observation is not mentioned in the abstract and the conclusions. Since this finding is important, it should be included in the discussion/conclusions sections to highlight the potential wider applicability of the approach.

To establish whether the observed synergistic effect is solely due to the increased uptake of the antibiotic component of the Trojan horse conjugate or rather the antimicrobial effect of the catechol-

bound Bi(III), a control compound consisting of a chloro-catechol amide (with no antibiotic) and its combination with Bi(III) should be investigated for comparison.

Abstract:

- the only clinically-used sideromycin that delivers...
- cope with the AMR crisis

Page 1:

Line 31: siderophores are

Line 36: there are already reports...

Line 38: novel strategies are urgently needed to...

Page 2:

Line 4: The role of metallo-sideromycins in the fight against AMR is discussed

Line 41: Since biofilm formation is one of the....

Page 3:

Line 22: To obtain further mechanistic insight into....

Line 57: This part of the sentence needs clarification.

Page 4:

Line 3: Please explain why carcinoma cells were chosen to examine the toxicity in mammalian cells. Healthy cells should be used in addition as a more representative example.

Page 5:

Line 31 ff: Please add the original data that underpin the results of the ICP-MS measurements to the SI.

Line 54 ff: A control experiment in which D₂O is titrated into the CEF solution is needed to establish if the decrease in the intensity of the catechol protons is not simply due to H/D exchange.

Table 1: B. cepacia

Manuscript: “Metallo-sideromycin: a double Trojan Horse strategy for combating antimicrobial resistance” (NCOMMS-22-46041A-Z)

Reviewer #1:

While the concept of binding non-iron metals to siderophores and sideromycins to generate novel antibiotics is not new, this detailed study of cefiderocol metal (notably bismuth) is noteworthy. As note, cefiderocol is the first marketed antibiotic containing an iron chelating group (chloro dihydroxy benzoate) that enables it to use siderophore mediated active transport processes to improve targeted delivery and activity against primarily Gram-negative bacteria.

Studies of sideromycins (siderophore-antibiotic conjugates) have revealed the tremendous potential of this so called “Trojan Horse” antibiotic approach. However, as most papers related to cefiderocol improperly state, the iron binding component of cefiderocol is NOT a siderophore but a mimic of the natural bis catechol components of real natural siderophores. Thus, throughout the paper, this should be clarified. Although now a bit older, still the best overall reference that should be included early in this paper (not later as reference 36) is that by Hider (Hider, R. C.; Kong, X. Chemistry and biology of siderophores. Nat. Prod. Rep. 2010, 27, 637–657). The appendices associated with that review provide wonderful sources of the structures of siderophores, their microbial producers and a wealth of addition information that is essential reading for anyone in the area. Some background is given to the sideromycin Trojan Horse strategy, but as also with most related papers, key references are missing. Over the last couple of decades, controversy has impeded the development of the potential of sideromycins (see reference 12 in the manuscript); however, this can unfortunately be attributed to the lack of reference to much earlier papers that demonstrate the clinical potential of natural sideromycins like albomycins. The authors reference albomycins via the Hider review (now ref 36), but do not include the essential references that demonstrate the efficacy in mammals, including thousands of patients. It is imperative that G. Benz, et al be given full credit for the first synthesis (which also determined the real structure) in the 1980s. Benz, G.; Schroeder, T.; Kurz, J.; Wuensche, C.; Karl, W.; Steffens, G.; Pfitzner, J. Konstitution der Desferriform der Albomycine $\delta 1$, $\delta 2$, ϵ . 1982, , 1322–1335. This pioneering and most impressive work must be referenced. The authors must include a reference to the extensive efforts by V. Braun and group related to studies of the albomycins and salmycins but should indicate that Braun demonstrated in vivo efficacy in animal studies: Biometals (2009) 22:3–13 DOI 10.1007/s10534-008-9199-7. Most importantly, the authors must include in the albomycin discussion the early studies by Gause on the discovery of albomycins and especially the report that they were used successfully in thousands of people in the Soviet Union in the late 1940s-50s and were remarkably effective in humans infected with already penicillin resistant bacteria. This work is seldom referenced, but attention to it would have avoided much of the controversy about the potential of sideromycins. This reference must be added along with related discussion: Gause GF (1955) Recent studies on albomycin, a new antibiotic. British Medical Journal 1776-1179.

Response:

We truly appreciate the review’s critical comments and helpful suggestions. We have clarified the iron binding component of cefiderocol is a mimic of the natural biscatechol components of natural siderophores, but not siderophore (Page 1, L44-45). We have also revised the introduction with suggested references cited and included the discovery and effectiveness of albomycins in humans infected with already penicillin resistant bacteria as suggested in the revised manuscript. (Page 1, L30-49).

We cited relevant references (references 9, 13, 14, 15 in the revised version). Related references cited are listed below:

Aside from the points made above, additional comments are provided below:

The authors state that they selected cefiderocol to showcase a series of metallo-compounds against different bacterial strains,

but this sounds overly broad when the focus is on gallium and primarily bismuth complexes of cefiderocol (though they did initially screen other metal complexes). The section on results provides good detail, including proper choice of media, (including iron sufficient and iron depleted media). They screened combinations of cefiderocol and bismuth and gallium salts against about 10 different classes of Gram-negative bacteria and only found an influence (32-64 reduction of MIC relative to cefiderocol itself) against the initial strains of *P. aeruginosa* and *B. cepacia*. The authors suggest that this selectivity might be due to the different bacterial strains having different iron requirements and uptake pathways, but this is curious since the other Gram-negative bacteria utilize catechol type siderophores. A better explanation is needed rather than the generalized speculation. With that initial determination of selectivity, the authors focused on *P. aeruginosa* strains and did proper controls by performing assays with cefiderocol alone, bismuth citrate alone and combinations in both iron sufficient and deficient media. The studies suggested that the combination of cefiderocol and bismuth gave improved activity to demonstrate their proposed “dual Trojan Horse” strategy.

Importantly, they also found that combination of cefiderocol and bismuth synergistically inhibited biofilm formation and suppressed resistance to cefiderocol in select strains of *P. aeruginosa*. Studies of isolated cefiderocol resistant strains indicated that combination of cefiderocol with bismuth resensitized the resistant strains to cefiderocol.

An additional important study was determination that bismuth binds to cefiderocol with 1:1 stoichiometry, consistent with the rather interesting and surprising claim by Shionogi that cefiderocol binds iron with 1:1 stoichiometry even though it contains only a single catechol unlike most siderophores or sideromycins. They also hypothesized that the bismuth complex would also suppress iron uptake and demonstrated consistent results using a relevant fluorescent probe. Also consistent with the active transport was the construction and study with a PAO1 mutant with a *piuA* gene (iron receptor) deletion.

Significantly, the authors demonstrated that the cefiderocol-bismuth complex was non toxic and somewhat effective in an in vivo study (an increase in survival rates to 75% compared to cefiderocol or bismuth citrate alone).

Overall, most of the results are consistent with the hypothesis that combination of cefiderocol and bismuth improves activity but only of select strains (which again is curious, since many of the bacteria studied and listed in Table 1, utilize catechols for iron transport). Further discussion of this selectivity is needed. Particularly, in most of the other strains in Table 1, the activity of the combination is essentially the same as for cefiderocol alone. It would be important to determine if in those cases, the bismuth conjugate was actually generated – the activity is consistent with the free non bismuth bound cefiderocol – why would it not form the bismuth complex in these cases? Also in Table 1, the activity of the other metal complexes is essentially the same as that for cefiderocol but it is well known that Chromium forms kinetically inert complexes with siderophores and, if cefiderocol also forms a chromium complex, it should have been recognized by the iron receptors and transporters and have been active like the bismuth complex. These results and the notable selectivity for *P. aeruginosa* need further study and elucidation.

If the changes suggested above are incorporated and IF the authors can provide data to rationalize the comments in the paragraph above, then the paper could be accepted in this high level journal. If not, I would recommend preliminary publication elsewhere. The manuscript should be screened for minor grammatical typographical errors that occur through the paper and distract from the science. It is not the role of a scientific reviewer to make such corrections and should be done by authors and the editorial office.

Response:

The synergy between CBS and CEF was observed only in *P. aeruginosa* and *B. cepacia* in the primary screening. In addition

to the fact that bacteria have different iron requirements and uptake pathways, one of the major reasons influencing the efficacy of metallo-sideromycins is that different bacteria strains showed different susceptibilities to Bi(III) drugs. Bi(III) drugs were generally considered as antimicrobial agents against *H. pylori*. CBS has also been shown to exhibit the best antibacterial activity against *P. aeruginosa* compared with *E. coli*, *S. aureus*, and *S. epidermidis*, and other strains (Vega-Jiménez, A.L. et al. *MRS Online Proceedings Library* 2012, **1487**, 14-18). In addition, P. Domenico *et al.* reported that combining Bi(III) salts with aminoglycosides or fourth-generation cephalosporin antibiotics may help to combat the resistant *P. aeruginosa* (Domenico, P. et al. *Eur J Clin Microbiol Infect Dis* 1992, **11**, 170-175). Our unpublished data also showed that bismuth complexes have a relatively low MIC (25 ug/ml) against *B. cepacia* strains. Similarly, Ga(III) has been used as antimicrobials to combat resistant *P. aeruginosa* (Wang, Y. et al. *Chem Sci* 2019, **10**, 6099-6106). However, the iron concentration severely affects the efficacy of Ga(III) compounds, and led a high MIC against *E. coli* under aerobic conditions, which may affect the synergy between Ga(III) and CEF (Neill, C.J. et al. *ACS Infect Dis* 2020, **6**, 2959-2969). Therefore, the combination of cefiderocol and bismuth/gallium showed enhanced activity only for certain strains, which is due to the selective antimicrobial activity of bismuth/gallium delivered to specific bacterial cells through cefiderocol siderophore pathways. To validate this hypothesis, we also measured the intracellular bismuth concentration in *E.coli* (as an example) after being co-treated with CBS and CEF. We found similar bismuth concentrations in both *E. coli* and *P. aeruginosa*, indicating that the similar amount of bismuth was transported into the bacterial cells and selective toxicity of CBS and CEF combination to different bacterial strains is attributable to selectivity of bismuth. These additional results and discussion have been incorporated into the revised version of the manuscript (Page 2 Line 29-35).

We agree with the reviewer that chromium(III) forms kinetically inert complexes with siderophores, and the Cr(III)-CEF may also be transported into bacterial cells through siderophore transporters. However, Cr(III) compounds showed less antibacterial activity against ESKAPE pathogens, thus no synergy between Cr(III) compounds and CEF was observed.

We also carefully read the manuscript and did our best to minimize grammatical typographical errors and those changes are highlighted.

Reviewer #2:

Sun and coworkers describe an investigation into the potential synergy of the sideromycin cefiderocol, a clinically approved antibiotic with metallodrugs/metal ions. The authors found that, sources of bismuth such as colloidal bismuth citrate see to effectively synergise with Cefiderocol in the treatment of *Pseudomonas aeruginosa*. This also translated somewhat to biofilm inhibition albeit without complete eradication. The binding of Bi(III) to cefiderocol was studied by standard techniques and it was shown that bacterial concentrations of bismuth increase in the presence of the sideromycin while iron concentrations decrease. Lastly the authors show that the cefiderocol-bismuth combination shows low toxicity in vitro and does show efficacy in an in vivo *P. aeruginosa* infection in mice. In this lung infection model the metal-sideromycin combination increased the survival rate and significantly reduced the bacterial load.

Overall this article by Sun and coworkers paints a promising picture for the usage of metal ions in conjunction with clinically approved antibiotics. The fact that both cefiderocol and CSB have been shown to be safe in humans would greatly accelerate the clinical development of such a combination. The one point of contention could be that this synergy seems to only apply to a fraction of the clinical *P. aeruginosa* population, so the applicability of this approach will have to be explored carefully. In the end I think this work presents noteworthy results worthy of publication in *Nature Communications*. There is an urgent need for new antibiotic strategies, particularly for Gram-negative ESKAPE pathogens, hence this work is of high significance. The

conclusions are supported by the experimental data provided.

Response:

We thank this reviewer's favorable comments and helpful suggestions for improving the manuscript.

There are a few comments that should be addressed before publication:

- The authors state that the combination of cefiderocol with CSB was tested against 60 clinical strains of *P. aeruginosa* but was only found to be synergistic in 40% of the cases which might reduce the clinical relevance of this treatment. Do the authors know why the synergy was not effective in 60% of these strains?

Response:

We randomly screened 60 clinical strains collected from the collection from our colleges in Medical School, and 24 showed synergy under the treatment of 16 μ M CBS. Unfortunately, we don't know the specific genotypes of these clinical strains and based on our data, these *P. aeruginosa* clinical strains are **not CEF resistant**, the combination of cefiderocol with CSB was found to be synergistic in 40% of the cases, which is already very good. We also included 2 clinical CEF-resistant *P. aeruginosa* strains, and the results show that in the presence of 32 μ M CBS, the MICs of CEF against both PA247 and PA245 were reduced to 1 μ M, suggesting that CBS could re-sensitize these resistant bacterial strains to CEF. These data are incorporated in the revised manuscript (Page 3, L42-45, Figure 2f, 2g).

- Against Biofilms the efficacy was found to be at maximum 80%. This seems suboptimal as I would assume a biofilm could simply regrow after treatment in such a scenario (was this observed)

Response:

We agree with this reviewer. Since we use the sub-inhibitory concentration to make sure the inhibition of biofilm is not due to the killing of bacterial cells, and the biofilm will be inhibited more than 92% compared to the control group if we treated with 16 μ M of CBS and 1 μ M of CEF.

- The NMR spectra given for the characterisation of the cefiderocol and bismuth are not ideal. Only a narrow area of the spectra is shown. The full NMR should be provided. Upon coordination of bismuth, one would expect a shift also in the protons belonging to the aromatic region of the catechol. Also, the authors use deuterated protonated solvents (D2O), this means the catechol protons could simply be exchanging with deuterium and hence disappear.

Response:

We thank for the suggestions. We have added the full NMR spectra in 90% H₂O and 10% DMSO-d₆ in the revised manuscript (Figure S9). And we apologize for the mistake in the description of the NMR titration experiment. We agree with the reviewer that coordination of bismuth to CEF induced the shift and change on the protons resonances belonging to the aromatic region of the 3-chlorocatechol. We have performed ¹H NMR titration experiments in H₂O and found the catechol protons are still not observable. However, we observed new peaks at 7.05 and 6.93 ppm appeared and increased in their intensity whereas the resonance at 6.82 and 6.73 ppm decreased their intensities and disappeared upon the addition of CBS, confirming that bismuth ions bind to the catechol and the free and bound CEF are in exchange on the ¹H NMR time scale.

- In the in vivo infection model cefiderocol alone seemed to not be effective at all in increasing survival, why is that? Should

it not be effective in principle?

Response:

We chose a relatively low concentration of cefiderocol which showed almost no effect in the animal model. And according to the reference, the MIC of cefiderocol in vivo is 5 mg/kg in the mice model (Katherine R. Hummels et al. *Eur. J. Med. Chem.* **155**, 847-868), and we only used 0.25 mg/kg, however, by combining this concentration of CEF with CBS, the infected mice could be rescued for 75%, which is very good indeed.

Reviewer #3:

This is an important manuscript that explores synergistic effects between the Trojan horse antimicrobial cefiderocol and Fe(III) mimics, in particular Bi(III). Interestingly, bismuth citrate was found to enhance the potency of cefiderocol against certain *P. aeruginosa* strains, both in vitro and in vivo. It is highly significant that the combination was found to impede biofilm formation and delay the development of resistance. The authors demonstrate that Bi(III) competes with Fe(III) by coordinating to the catechol-part of cefiderocol and uptake studies suggest that the resulting Bi(III) complex is taken up via the iron-siderophore pathway. This is a potentially transformative development. Considering the increasing threat of antimicrobial resistance, the manuscript merits publication in Nature Communications.

There are, however, a number of points to be addressed before publication can be considered:

General points:

It was found that both Bi(III) and Ga(III) are acting synergistically with cefiderocol, however, only Bi(III) is mentioned in the abstract and in the conclusion section. Similarly, only colloidal bismuth citrate (CBS) was progressed to biofilm formation, resistance evolution, cellular uptake and in vivo studies. The authors should explain in the text why Ga(III) and the other Bi(III) salts that showed promise in the initial activity screens were not investigated further.

Response:

We thank this reviewer for favorable comments and helpful suggestions. We did not carry out exact the same studies on gallium compared with bismuth. We only used bismuth as a showcase study and will definitely perform very detail investigation on other bismuth compounds and Ga(III) in the future as suggested. We indicated clearly that Bi(III) drug e.g. CBS, was selected as a showcase study in the revised manuscript (Page 2 Line 49).

Synergism was also observed with certain *B. cepacia* strains. However, this observation is not mentioned in the abstract and the conclusions. Since this finding is important, it should be included in the discussion/conclusions sections to highlight the potential wider applicability of the approach.

Response:

We thank for the suggestions, and revised the abstract and the conclusions to include the observation of *B. cepacia* strains as suggested.

To establish whether the observed synergistic effect is solely due to the increased uptake of the antibiotic component of the Trojan horse conjugate or rather the antimicrobial effect of the catechol-bound Bi(III), a control compound consisting of a

chloro-catechol amide (with no antibiotic) and its combination with Bi(III) should be investigated for comparison.

Response:

We agree with this reviewer and thank for the suggestions. We investigated the combination of 3-chlorocatechol with Bi(III) against PAO1 and did not find any synergy between them. These data are included in the revised manuscript (Figure S2).

We thank this reviewer's careful reading and have corrected the errors raised by the reviewer.

Abstract:

- the only clinically-used sideromycin that delivers...
- cope with the AMR crisis

Page 1:

Line 31: siderophores are

Line 36: there are already reports...

Line 38: novel strategies are urgently needed to...

Page 2:

Line 4: The role of metallo-sideromycins in the fight against AMR is discussed

Line 41: Since biofilm formation is one of the....

Page 3:

Line 22: To obtain further mechanistic insight into....

Line 57: This part of the sentence needs clarification.

Response:

We thank the reviewer for the suggestion and we have revised the sentences (Page 4, L27-28).

Page 4:

Line 3: Please explain why carcinoma cells were chosen to examine the toxicity in mammalian cells. Healthy cells should be used in addition as a more representative example.

Response:

We thank the reviewer for the suggestions, and we've examined the toxicity in the healthy cells embryonic kidney cells (HEK293T) and included this in the revised manuscript (Page 4, L33, Figure S11).

Page 5:

Line 31 ff: Please add the original data that underpin the results of the ICP-MS measurements to the SI.

Line 54 ff: A control experiment in which D2O is titrated into the CEF solution is needed to establish if the decrease in the intensity of the catechol protons is not simply due to H/D exchange.

Response:

We have added the original data of ICP-MS in Table S4.

We thank for this review's suggestion. We apologize for not clearly stated that the NMR study was studied in D₂O originally and reexamined in 90% H₂O in the revised manuscript. In both cases, the 3-chloro-catechol protons could not be observed. Only the protons δ 6.82 (1H, d) and 6.73 (1H, d) could be detected, which are assignable to the protons from the catechol (protons **1#** and **2#**). The peaks decreased their intensities and gradually disappeared owing to the binding to CBS and new peaks also appeared and broadened at 7.05 and 6.93.

Table 1: B. cepacia

Response:

This has been corrected.

REVIEWER COMMENTS

Reviewer #1 (Remarks to the Author):

The authors have addressed most of the points I raised as reviewer #1.

I would suggest just adding a bit to the important sentence about Gause's work with albomycin to indicate that the success was noted in people:

For example, albomycins were first obtained from *Streptomyces* strains¹³, characterized by G. Benz et al. ¹⁴ in 1982, and used successfully IN PEOPLE in the late 1940s against penicillin-resistant bacteria.

Based on the other reviews, there might still be concerns about the NMR data. Even though they now took NMR spectra in H₂O instead of D₂O, there should be rapid exchange between the catechol OH groups and water to give an OH shift - is that all that is being seen rather than confirming binding? I will let the other reviewers address this since they initially pointed out the concern.

Reviewer #2 (Remarks to the Author):

I want to thank the authors for their efforts to address the concerns of myself and the other reviewers. I am satisfied with how my comments were addressed in general.

However the ¹H-NMRs provided in Figure S9 are a bit strange and seem to not have enough peaks for CEF. There are several CH₂s and other peaks missing so I'm not sure what happened there. Also it is not clear to me why the aromatic catechol signals (1 and 2) would disappear after binding of CBS?

Reviewer #3 (Remarks to the Author):

All my comments have been addressed appropriately and the manuscript is now suitable for publication.

Manuscript: “Metallo-sideromycin: a double Trojan Horse strategy for combating antimicrobial resistance” (NCOMMS-22-46041A-Z)

Reviewer #1:

The authors have addressed most of the points I raised as reviewer #1.

I would suggest just adding a bit to the important sentence about Gause's work with albomycin to indicate that the success was noted in people:

For example, albomycins were first obtained from *Streptomyces* strains¹³, characterized by G. Benz et al. 14 in 1982, and used successfully IN PEOPLE in the late 1940s against penicillin-resistant bacteria.

Response:

We thank this reviewer’s helpful suggestions. We have revised the introduction about Gause’s work as suggested.

Based on the other reviews, there might still be concerns about the NMR data. Even though they now took NMR spectra in H₂O instead of D₂O, there should be rapid exchange between the catechol OH groups and water to give an OH shift - is that all that is being seen rather than confirming binding? I will let the other reviewers address this since they initially pointed out the concern.

Response:

We have performed the ¹H NMR experiment in 90% H₂O to see if the active protons at OH groups can be observed in the NMR spectrum as suggested, and unfortunately the catechol -OH group still cannot be observed possibly due to the exchange. However, we did observe the changes of aromatic proton signals upon the titration of Bi(III), attributable to the binding of Bi(III) (as CBS) to CEF, which changes the chemical environment of the aromatic protons, thus resulting in the changes of the signals. However, the free and bound forms of these resonances are exchanged on ¹H NMR time-scale, i.e. slow, intermediate and fast (Macomber RS. *J Chem Edu* 1992, **69**, 375), leading to the change of NMR resonances differently. For example, both the bound and free forms can be observed for slow exchange, i.e. the free form decreases in intensity and bound form increases. For the bound and free forms are in fast exchange, one observes an averaged signal shifting from the free to the bound form. For those in intermediate exchange, one will observe broadening and even disappearing of the NMR resonances. In this work, the resonances are broadened and even disappeared upon addition of Bi(III) to the CEF solution, indicative of an intermediate exchange (Kleckner IR *et al. Biochim Biophys Acta*. 2011, **1814**, 942). To further confirm this, we redid the experiments at higher temperature (i.e. 37 °C), and observed that the broadened resonances in the ¹H NMR spectrum became sharper owing to increased exchange rate.

Reviewer #2:

I want to thank the authors for their efforts to address the concerns of myself and the other reviewers. I am satisfied with how my comments were addressed in general.

However the ¹H-NMRs provided in Figure S9 are a bit strange and seem to not have enough peaks for

CEF. There are several CH₂s and other peaks missing so I'm not sure what happened there.

Response:

We appreciate this reviewer's favorable comments and suggestions. The ¹H NMR spectra shown in the current manuscript were recorded in 90% H₂O, and a WATERGATE pulse sequence was used to suppress the water resonance. Those resonances (i.e. peaks) of CEF in 3.8-5.5 ppm might be affected by the water suppression technique, and disappeared in the spectrum. We also acquired ¹H NMR spectra in D₂O (as shown below) under identical conditions and these resonances could be observed clearly. Nevertheless, binding of CBS to CEF did not result in any changes of these resonances. Instead, the proton resonances in the catechol (chlorocatechol, i.e. 6.9-7.1 ppm) broadened and disappeared upon addition of CBS, which is the same as the NMR titration results in 90% H₂O.

Also it is not clear to me why the aromatic catechol signals (1 and 2) would disappear after binding of CBS?

Response:

The disappearance of the aromatic catechol resonances (i.e. protons **1** and **2**) upon binding of CBS is due to the intermediate exchange of the bound- and free-forms of CEF on the ^1H NMR time scale. Detailed description of slow exchange, fast exchange and intermediate exchange respectively on the NMR time scale was described previously (Macomber RS. *J Chem Edu* 1992, **69**, 375). In this manuscript, the resonances of **1** and **2** in the catechol are broadened and even disappeared upon Bi(III) binding, indicative of a typical intermediate exchange (Kleckner IR *et al. Biochim Biophys Acta*. 2011, **1814**, 942). As shown below, when we raised the temperature to 37 °C, the broadened resonances became sharper and slightly shifted (owing to different temperature), confirming the disappearance of the peaks **1**, **2** is owing to the binding of CBS to CEF, and the bound- and free-forms are in an intermediate exchange in the ^1H NMR time scale.

Reviewer #3:

All my comments have been addressed appropriately and the manuscript is now suitable for publication.

Response:

We thank this reviewer's favorable comments and helpful suggestions for improving the manuscript.

REVIEWERS' COMMENTS

Reviewer #2 (Remarks to the Author):

I thank the authors for their work. I believe if the supplemented NMR spectra are added to the SI of the manuscript it is now suitable for publication.

**Manuscript: “Metallo-sideromycin: a double Trojan Horse strategy for combating antimicrobial resistance”
(NCOMMS-22-46041A-Z)**

Reviewer #2:

I thank the authors for their work. I believe if the supplemented NMR spectra are added to the SI of the manuscript it is now suitable for publication.

Response:

We thank this reviewer’s favorable comments and helpful suggestions for improving the manuscript. We have added the supplemented NMR spectra to the Supporting Information.